# Antiretroviral Treatment-Induced Galectin-9 Might Impact HIV Viremia in Addition to Contributing to Inflammaging

**DOI:** 10.3390/ijms241512273

**Published:** 2023-07-31

**Authors:** Ashwini Shete, Vaishnav Wagh, Jyoti Sawant, Pallavi Shidhaye, Suvarna Sane, Amrita Rao, Smita Kulkarni, Manisha Ghate

**Affiliations:** Indian Council of Medical Research, National AIDS Research Institute (ICMR-NARI), Pune 411026, India; vaishnavwagh04@gmail.com (V.W.); sawantjyoti11@gmail.com (J.S.); pshidhaye@nariindia.org (P.S.); arao@nariindia.org (A.R.); skulkarni@nariindia.org (S.K.); mghate@nariindia.org (M.G.)

**Keywords:** Galectin-9, cystatin C, antiretroviral therapy, HIV viremia, inflammaging

## Abstract

Background: Galectin-9 induces HIV reactivation and also contributes to non-AIDS events through inflammaging. Hence, it is important to assess its levels in HIV-infected individuals to determine their association with HIV viremia and other comorbidities. Methods: Plasma galectin-9 levels were estimated in viremic *(n* = 152) and aviremic (*n* = 395) individuals on first-line antiretroviral therapy (ART). They were assessed for correlation with HIV-1 viral load (VL), CD4 count, and ART duration, as well as for receiver operating characteristic curve analysis. Result: Plasma galectin-9 levels correlated positively with VL (r = 0.507, *p* < 0.0001) and ART duration (r = 0.308, *p* = 0.002) and negatively with CD4 count (r = −0.186, *p* < 0.0001). Area under the curve for galectin-9/CD4 count ratio for identifying viremic individuals was 0.906. Sensitivity and specificity of the ratio at a cutoff of 14.47 were 90.13% and 70.05%, respectively, for detecting viremic individuals. Further, galectin-9 levels correlated with cystatin C (r = 0.239, *p* = 0.0183), IL-18 (r = 0.311, *p* = 0.006), and systolic blood pressure (r = 0.220, *p* = 0.0355). Galectin-9-induced HIV reactivation was significantly lower in individuals on long-term ART than those on short-term ART. Conclusion: The galectin-9-to-CD4 count ratio indicated the potential of galectin-9 as a cheaper monitoring tool to detect HIV viremia. Strategies for countering the effects of galectin-9 for controlling HIV viremia and non-AIDS events are urgently warranted.

## 1. Introduction

Galectin-9 is an immunomodulatory protein belonging to the family of B-galactosidase binding lectins. It promotes immunosuppressive and proinflammatory responses based on the phases of the host immune responses [1]. It induces inflammation and T-cell apoptosis and modulates HIV infection through receptors such as T-cell immunoglobulin mucin domain-3 (Tim-3) and protein disulfide isomerases (PDI) [2]. It is a tandem repeat-type galectin with an N-terminal carbohydrate recognition domain (CRD) and a C-terminal CRD linked together by a linker domain. N-terminal and C-terminal CRDs (NCRD and CCRD) of galectin-9 have only 39% amino acid homology, unlike Galectin-1, where two CRDs are identical [3]. Owing to the difference in the CRD sequences, their binding specificities also differ. This results in differential functionalities mediated by them by acting through various receptors as well as by targeting different cells [4,5]. We previously showed the potential of NCRD in inducing Treg cells, whereas CCRD was shown to inhibit regulatory responses [6]. Galectin-9 is susceptible to proteolytic breakdown by extracellular proteases, which are increased in inflammatory conditions prevalent during HIV infection, generating free NCRDs and CCRDs. This type of galectin-9 is called cleaved galectin-9. Studies have shown that total plasma galectin-9 consisting of free NCRD, CCRD, and full-length Gal-9 levels are high in HIV patients with virologic failure [7,8].

Reports suggest that human Gal-9 regulates p21 expression, indicating its possible role in HIV transcription and latency reversal [9]. In spite of having structural similarities, other galectins, such as galectin-4 and galectin-8, do not reactivate latent HIV infection [9]. Full-length and individual CRDs of Gal-9 have the potential to reverse HIV latency in vitro and ex vivo in a dose-dependent manner [6,10]. This is possibly the reason for the positive association between total galectin-9 levels and plasma HIV viral load in a couple of studies performed elsewhere. Hence, galectin-9 levels have been identified as a cheaper surrogate marker of HIV viremia in individuals diagnosed with HIV and on antiretroviral therapy (ART) but not virally suppressed. HIV viral load testing is a crucial component of ART monitoring in the management of HIV-infected individuals. However, there is still a scarcity of medical infrastructure in resource-constrained countries that greatly affects the periodic monitoring of HIV viremia, leading to high prevalence and mortality rates of AIDS even in the era of antiretroviral therapy. With the UNAIDS 95-95-95 target, there is a global expansion of people living with HIV (PLHIV) on ART, which has further increased demand for VL monitoring. Hence, there are still gaps in the coverage of VL testing in low- and middle-income countries (LMICs), which ranges from 12 to 96%, indicating the need for using different strategies to reach the set goals [11]. Point-of-care tests and cheaper biomarkers serving as surrogate markers of viremia might help in detecting virologically unsuppressed individuals in the absence of availability of viral load monitoring. An immunological profile comprising high CD4+CD38+ cells and high plasma levels of soluble CD14 and soluble endothelial protein C receptor have been found to correlate with residual viremia in one study [12].

We showed a strong correlation of plasma galectin-9 with HIV viral load in PLHIV on ART for a one-year duration, possibly due to its effect on HIV reactivation [7]. However, data of PLHIV on ART for a longer duration of time have not been studied previously. Apart from inducing HIV reactivation, galectin-9 levels have also been associated with occurrence of non-AIDS events in HIV-infected individuals [13]. It has been shown to act as a promoter of atherosclerosis [14] and has been linked to several age-related complications in HIV-uninfected individuals [15,16,17]. Its association with multiple inflammatory markers indicates its possible role in inflammaging [18]. Since HIV-infected individuals are at a higher risk of premature aging [19], it is important to assess levels of galectin-9 in HIV-infected individuals to understand their role in age-related morbidities in addition to inducing HIV viremia. Galectin-9 levels were shown to be positively correlated with cystatin C levels in one of the cohort studies, the aim of which was to determine the prevalence of age-related comorbidities in people living with HIV with long-term viral suppression [20]. The study also showed a direct correlation of these markers with the total number of comorbid conditions in the same cohort [20]. Cystatin C has also been shown to affect HIV replication [21] and might affect HIV viremia in PLHIV.

Hence, we planned a study to assess plasma galectin-9 levels in PLHIV on ART and their associations with plasma viral load, cystatin C levels, and other parameters possibly contributing to non-AIDS events. We assessed whether the plasma total galectin-9 levels are a more sensitive marker of HIV viremia than the full-length galectin-9 levels, as cleaved galectin-9 CRDs also contribute to inducing HIV reactivation.

## 2. Results

### 2.1. Characteristics of Study Participants Enrolled in the Study

Characteristics of study participants from viremic and aviremic groups are mentioned in Table 1. A total of 152 viremic individuals (male:female—76:76) with a median age of 43.5 years and 395 aviremic individuals (male:female:TG—174:220:1) with a median age of 43 years were enrolled in the study. The median CD4 count for the viremic PLHIV was 259.5 cells/cmm (IQR: 167–402) and that of aviremic PLHIV was 547 cells/cmm (IQR: 388–771). The median VL for viremic PLHIV was 15,392 (IQR: 3373–57,455) copies/mL. The median duration of ART was 3 and 4 years in viremic and aviremic PLHIV, respectively. These two groups did not differ significantly from each other with respect to age, sex, and duration of ART. The total galectin-9 levels were significantly higher (*p* < 0.0001) in viremic PLHIV (median: 10,712; IQR: 7558–13,699 pg/mL) than those in aviremic PLHIV (median: 5768; IQR: 3313–8689) pg/mL. The median of total galectin-9 levels in normal HIV-uninfected individuals (median age—47 years; male:female—32:37) was 3407 (IQR: 2163–4991) pg/mL, which was significantly lower compared to viremic and aviremic PLHIV (*p* < 0.0001). The full-length galectin-9 levels did not differ significantly among viremic (median: 1365; IQR: 409–3266) pg/mL and aviremic PLHIV (median: 1257; IQR: 425–3193) pg/mL.

### 2.2. Correlation of Galectin-9 Levels with Different Parameters

Although full-length galectin-9 levels did not differ between viremic and aviremic individuals, they correlated positively with total galectin-9 levels (r = 0.2829, *p* = 0.0013). Plasma total galectin-9 values correlated positively (r = 0.5073, *p* < 0.0001) with VL values. Further correlation analysis of total galectin-9 levels was also carried out with CD4 count. The levels showed a negative correlation with absolute and %CD4 count in the study participants (r = −0.1863, *p* < 0.0001 and r = −0.2592, *p* < 0.0001, respectively). Total galectin-9 levels also showed a weak correlation with age (r = 0.0937, *p* = 0.0283). Interestingly, total galectin-9 levels and the full-length-to-total galectin-9 ratio showed a positive correlation with the duration of ART (r = 0.2278, *p* < 0.0001 and r = 0.323, *p* = 0.003, respectively).

Since total galectin-9 levels were observed to increase with the duration of ART, the study participants were further split into two groups: those on ART for up to 2 years and those on ART for more than 2 years. Total galectin-9 levels correlated more strongly with VL in PLHIV on ART for up to 2 years’ duration (r = 0.665, *p* < 0.0001) than those on ART for more than 2 years, as shown in Figure 1. The correlation was poor in PLHIV on ART for more than 2 years (r = 0.410, *p* < 0.0001). The correlation with CD4 count was significant but weak in PLHIV on ART for 2 years and for more than 2 years (r = −0.233, *p* = 0.0004 and r = −0.251, *p* < 0.0001, respectively).

VL values were found to correlate negatively with absolute and %CD4 values (r = −0.5064, *p* < 0.0001 and r = −0.34, *p* < 0.0001, respectively), as shown in Table 2. The VL also correlated negatively with the duration of ART, although the correlation was very weak (r = −0.0915, *p* = 0.0323). The VL correlated more strongly with the total galectin-9-to-CD4 count ratio than with the individual parameters (r = 0.6422, *p* < 0.0001).

### 2.3. Receiver Operating Characteristic (ROC) Curves of Galectin-9 and VL for Monitoring HIV-1 Viremia in PLHIV with Virologic Failure

Since VL correlated positively with total galectin-9, ROC analysis was carried out to determine the cutoff of total galectin-9 (Figure 2). The area under the ROC curve for total galecin-9 was 0.813, which determined a cutoff of 8.06 ng/mL of total galectin-9 levels to differentiate viremic individuals from aviremic individuals, with a sensitivity and specificity of 69.47% and 70.13%, respectively. Since a sensitivity of total galectin-9 levels for identifying viremic individuals was found to be below 70% at a specificity of 70%, ROC analysis was also performed for the ratios of total galectin-9 levels to CD4 count and to duration on ART in years. It was observed that the total galectin-9 levels correlated with the CD4 count and years after ART initiation. Hence, those ratios were used for ROC analysis. The area under the curve (AUC) for the ratio of total galectin-9 to CD4 count in PLHIV on ART was 0.9063, whereas that for the ratio of total galectin-9 levels to years on ART was 0.719. Median galectin-9-to-CD4 cell ratios were 36.96 (IQR: 22.53–78.23), 9.95 (IQR: 6.06–16.46), and 3.97 (IQR: 2.46–7.31) in viremic HIV-infected, aviremic HIV-infected, and HIV-uninfected individuals, respectively. The cutoff value of 14.47 of the total galectin-9/CD4 count ratio was observed to differentiate viremic individuals from aviremic ones with a sensitivity of 90.13% and a specificity of 70.05%. The cutoff identified showed 90% sensitivity with 15 viremic individuals yielding a ratio below the cutoff. Thirteen out of these 15 missed individuals had a VL below 5000 copies/mL.

### 2.4. Plasma Galectin-9 Levels Correlate with Cystatin C Levels in Aviremic PLHIV on Long-Term ART

The correlation of galectin-9 levels with other parameters like cystatin C levels and blood pressure was also analyzed in a subset of aviremic individuals on long-term ART to determine their association with factors contributing to non-AIDS events (Table 2). The galectin-9 levels correlated positively with systolic blood pressure (r = 0.220, *p* = 0.0355, one-tailed) and plasma cystatin C levels (r = 0.239, *p* = 0.0183). Total galectin-9 levels were also assessed for their correlation with inflammatory markers like IL-1β, IL-6, and IL-18. The levels did not show any correlation with IL-1β (r = −0.024, *p* = 0.425) and IL-6 (r = 0.003, *p* = 0.489). However, positive correlation of the levels with IL-18 (r = 0.311, *p* = 0.006) was observed. Further, cystatin C levels correlated with the age and duration of ART in aviremic PLHIV on long-term ART (r = 0.209, *p* = 0.04 and r = 0.308, *p* = 0.002, respectively). Cystatin C levels were also analyzed to determine their correlation with renal function tests to assess whether their levels were associated with renal dysfunction. It was observed that the plasma cystatin C levels correlated positively with serum creatinine (r = 0.3277, *p* = 0.0053) and blood urea nitrogen levels (r = 0.276, *p* = 0.0206), as shown in Table 2. However, none of the patients presented with overt renal dysfunction.

### 2.5. HIV Reactivation in PBMCs of HIV-Infected Individuals on Short-Term vs. Long-Term ART after Stimulation with Recombinant Galectin-9

Recombinant galectin-9 induced HIV reactivation after treatment in PBMCs of HIV-infected individuals on short-term and long-term ART as shown in Figure 3. The expression index of the *gag* gene was significantly higher after galectin-9 treatment in individuals on short-term ART compared to the unstimulated controls (*p* = 0.0273). It was higher but not significant in those on long-term ART (*p* = 0.0547). Intracellular P24 expression in CD4+ T cells was significantly upregulated in individuals on long- and short-term ART (*p* = 0.0078 and 0.0029, respectively) after stimulation with recombinant Gal-9. The expression index (*p* = 0.0037 and 0.0016, respectively) and P24-expressing CD4+ T cells (*p* = 0.06 and 0.05, respectively—not significant) were lower in those on long-term ART than in those on short-term ART before and after treatment with galectin-9. Further, we compared the expression of Tim-3, which acts as a ligand for galectin-9, on CD4+ and CD8+ T cells of these individuals. Interestingly, there was a significant upregulation of Tim-3 expression on both CD4 and CD8+ T cells in individuals on long-term ART compared to those on short-term ART (*p*-value < 0.0001 for both CD4 and CD8+ T cells).

## 3. Discussion

Galectin-9 plays an important role in many inflammatory and infectious diseases [17,22,23]. The role of plasma galectin-9 levels as a prognostic marker for monitoring HIV viremia in individuals diagnosed with HIV infection and on ART for one year had been reported by us and by others previously [7,8]. In this study, we investigated the role of plasma galectin-9 in differentiating viremic and aviremic individuals on ART for longer durations. Experiments encompassed quantifying plasma galectin-9 and examining the relationship between galectin-9 and years on ART, markers of disease progression, and cystatin C levels. We observed increased plasma total galectin-9 levels in viremic PLHIV on ART for varying durations compared to in aviremic individuals, as also reported by us previously [7]. Plasma total galectin-9 levels correlated positively with the HIV-1 VL and negatively with CD4 count, indicating their association with HIV disease progression. The correlation of the levels with HIV-1 VL was stronger in PLHIV on ART for up to 2 years than in those on ART for more than 2 years. The levels were found to correlate positively with duration of ART even in aviremic individuals, suggesting the involvement of factors other than HIV viremia playing a role in its induction and vice versa, weakening their association with plasma viremia and CD4 count. Differing associations of the levels with CD4 count have been reported, from −0.58 in pre-ART individuals [13] to no correlation where individuals on ART were grouped [2].

ROC curve analysis was further used to determine the discriminatory potential of total galectin-9 in identifying viremic individuals on ART. The area under the curve (AUC) was 0.813, with around 70% sensitivity for a specificity of 70% at the cutoff of 8.06 ng/mL, indicating low sensitivity of the levels to discriminate between viremic and aviremic individuals if used alone. The AUC and sensitivity were observed to be lower than reported by us in individuals on ART for a one-year duration, precluding utility of the levels as a surrogate marker of HIV viremia when used alone. Since CD4 count also correlated negatively with total galectin-9 levels and VL, we performed ROC analysis using a ratio of total galectin-9 levels to CD4 count for identifying viremic individuals on ART. The area under curve for the ratio was 0.906, with around 90% sensitivity for a specificity of 70% at a cutoff value of 14.47. This indicated a role of the ratio of total galectin-9 levels to CD4 count as a screening test to identify viremic PLHIV on ART for longer durations. Around 87% of the missed viremic PLHIV had an HIV-1 VL of less than 5000 copies/mL, indicating a sensitivity of more than 98% in identifying viremic individuals with VL values above 5000 copies/mL. However, the specificity of 70% indicated a necessity for a confirmatory VL test to rule out its increase due to other causes of immune activation.

Galectin-9 protein has been shown to enhance HIV transcription through TCR-dependent ERK-based pathways [10], which is likely to be the reason for its association with plasma viremia. We observed a correlation of total galectin-9 levels compared to full-length galectin-9 levels with plasma viremia. Full-length levels are likely to indicate synthesized galectin-9 levels, and total galectin-9 levels reflect the synthesized fraction and the fraction that has undergone enzymatic degradation. Viremic PLHIV are likely to have increased immune activation and ongoing inflammation due to higher VL, which might be responsible for increased cleavage of full-length galectin-9, contributing to higher total galectin-9 levels. We have reported that the individual N- and C-terminal carbohydrate-binding domains of galectin-9, which are likely to be formed after enzymatic cleavage of the inker region, are able to Induce HIV transcription in addition to the full-length galectin-9 protein [6]. Hence, total galectin-9 levels might have been observed to strongly correlate with HIV-1 viremia and have the potential to serve as a surrogate marker of HIV viremia compared to full-length galectin-9 protein.

Total galectin-9 levels were found to correlate positively with duration of ART in aviremic individuals, indicating the possible role of antiretroviral drugs in inducing it. The full-length-to-total galectin-9 ratio also showed a direct correlation with duration of ART, indicating the presence of a higher proportion of full-length galectin-9 than its cleaved forms with long-term virally suppressive ART. A higher proportion of full-length galectin-9 might indicate increased synthesis and decreased cleavage of galectin-9 in these individuals. Contrarily, viremic individuals are more likely to have higher levels of immune activation and inflammatory responses, leading to more cleavage of the protein by various inflammatory enzymes. The cleavage might lead to higher total galectin-9 levels and reduced full-length galectin-9 levels, leading to differential associations observed between these levels and plasma viremia. Cleaved galectin-9 has been shown to be a better biomarker of inflammation and severity in AIDS than the full-length protein [24]. In spite of the increased galectin-9 levels in PLHIV on long-term ART, they were able to maintain their aviremic status. Hence, we assessed HIV reactivation after stimulation of PBMCs with recombinant galectin-9 in virally suppressed individuals of ART for short-term duration versus those on long-term duration. Intracellular P24 expression, indicative of HIV reactivation, increased significantly after stimulation with exogenous galectin-9 in individuals on short-term and long-term ART. However, cells expressing *gag* copies and P24 were significantly lower in individuals on long-term ART than in those on short-term ART, indicating a reduction in their circulating viral reservoir. Virally suppressive ART is responsible for the decay of the cellular reservoir, leading to a lower percentage of P24-expressing CD4 cells after treatment with anti-latency agents, as also reported earlier [25]. HIV reactivation detected in our study was minimal, as the participants were on virally suppressive ART. We did not find much reactivation in ex vivo assays performed previously using PBMCs of virally suppressed PLHIV on ART after stimulation with positive controls like antiCD3/CD28 antibodies and phorbol myristate acetate /ionomycin. Hence, we did not use these controls to save on PBMCs, which were required to be used for flow and real-time assays. We also assessed whether duration of ART decreased the expression of the galectin-9 ligand Tim-3 on CD4+ and CD8+ T cells to determine whether it contributes to reduced reactivation in those on long-term ART. Interestingly, Tim-3 expression increased significantly in those on ART for long duration, as also reported in another study [26]. Galectin-9–Tim-3 interaction has been shown to provide resistance against infection with HIV-1 [27], which might play a beneficial role in terms of HIV-1 control in individuals on long-term ART.

Aviremic status in individuals on long-term ART despite the increased galectin-9 levels might also indicate the presence of some countering mechanism. Human cystatin C is a thiol proteinase inhibitor that exhibits antiviral activities against herpes simplex virus (HSV) [28,29], human coronaviruses [30], and HIV by influencing the maturation process [21,31]. Moreover, cystatin C fragments were shown to inhibit GPR15-mediated HIV infection [32]. Thus, contrary to galectin-9, cystatin C has been shown to possess inhibitory activities against HIV in different studies. We found that cystatin C levels that correlated positively with total galectin-9 levels in aviremic individuals might counter the effect of galectin-9 on HIV replication. However, the effect of the combination of galectin-9 and cystatin C on HIV reactivation needs to be further investigated to ascertain the association.

Plasma cystatin C levels also correlated positively with the duration of ART in our study participants. Cystatin C has emerged as an early marker of renal dysfunction [33], and various antiretroviral drugs have shown to impair kidney function [34]. We observed that serum creatinine and blood urea nitrogen levels in these patients correlated positively with plasma cystatin C levels, although none of the patients enrolled in our study presented with renal dysfunction. Apart from renal dysfunction, cystatin C has been proposed to be a predictor of cardiovascular disease (CVD) and all-cause mortality, including that due to CVD [35,36]. Similar to cystatin C, plasma galectin-9 has also been shown to be associated with aging-related comorbidities such as type 2 diabetes, atherosclerotic stroke, and coronary artery disease in various studies [15,16,17]. Both levels positively correlated with the age of the participants in our study. Positive associations between cystatin C and galectin-9 levels and age have been reported in healthy and/or diabetic individuals [16,37]. Both cystatin C and galectin-9 have been shown to be associated with various non-AIDS adverse events in persons with chronic HIV during suppressive antiretroviral therapy [13,38]. We could not assess the occurrence of non-AIDS events in our study, as it was cross-sectional in nature. However, we found a positive association of systolic blood pressure with plasma galectin-9 values in our study. ART has been shown to elevate blood pressure, increasing cardiovascular risk [39]. The risk of atherosclerotic cardiovascular disease is shown to increase with increasing systolic blood pressure even in individuals without hypertension [40], and hence, targeting systolic blood pressure is very important to reducing cardiovascular risk [41]. Most non-AIDS morbidities are linked to inflammaging being responsible for premature aging in PLHIV [42,43]. The association of galectin-9 levels with inflammatory markers was assessed to determine their possible contribution to the inflammaging process. Total galectin-9 levels were found to correlate with plasma IL-18 levels, as also reported in one of the inflammatory diseases [44]. IL-18 has been shown to participate in processes involved in inflammaging [45], and increased levels of IL-18 have been reported in individuals with metabolic syndrome [46]. Levels of both cystatin C and galectin-9 correlating with duration of ART suggest a need for screening PLHIV on long-term ART for non-AIDS events.

Limitations of our study include its cross-sectional design, where levels were determined only during one visit. It would be good to measure serial levels at different time points after ART initiation to confirm their association with ART duration. We also did not follow up with these patients to assess the occurrence of non-AIDS events in them and to determine their association with galectin-9 levels. Participants enrolled in our study differed from each other with respect to demographic characteristics and clinical status, which might have influenced the levels. Multiple mechanisms are involved in the development of AIDS-related as well as non-AIDS events, apart from galectin-9. Hence, correlations of galectin-9 levels with multiple parameters reported in our study were weak. Nonetheless, these statistically significant correlations do point to the involvement of the levels in mediating HIV viremia and inflammaging in these patients, indicating a need to target them.

## 4. Materials and Methods

### 4.1. Study Population

The study was conducted at the ICMR-National AIDS Research Institute (ICMR-NARI), India, after the approval of the designed protocol by the ICMR-National AIDS Research Institute Ethics Committee. Written informed consent was obtained from the study participants before enrolling them in the study. Both viremic (>1000 copies/mL) and aviremic PLHIV on antiretroviral therapy were enrolled from the institute’s clinic. Viremic and aviremic HIV PLHIV were categorized based on VL monitoring by Abbott real-time HIV-1 viral load assay (Abbott, Chicago, IL, USA) with a lower limit of detection of 40 RNA copies/mL. Participants with a VL of less than 40 copies/mL or target not detected (TND) were selected as aviremic, whereas those with a VL of more than 1000 copies/mL were selected as viremic PLHIV. Assessment of serum creatinine, blood urea nitrogen, blood pressure, and inflammatory markers was completed on a subset of aviremic individuals (*n* = 70). HIV-uninfected individuals (*n* = 69) were also enrolled to assess total galectin-9 levels in them.

### 4.2. Total Galectin-9 ELISA Assay

Plasma concentrations of galectin-9 were measured using the Human Galectin-9 DuoSet ELISA Kit (Cat# DY2045, R &D Systems, Minneapolis, MN, USA), which detects full-length and cleaved products of galectin-9, as it uses both capture and detection antibodies directed towards N-CRD of galectin-9. The ELISAs were performed by strictly abiding by the protocols in the manufacturer’s manual.

In brief, 96-well plates from an ancillary kit (Cat# DY008, R &D Systems, Minneapolis, MN, USA) were coated with mouse anti-human galectin-9 capture antibody (mAb), blocked with 1% BSA in PBS (reagent diluent), then incubated for 1 h at 37 °C with 1:10-fold diluted plasma samples. After several washes, galectin-9 in the wells was recognized by biotinylated goat anti-human galectin-9 antibody. Quantification was performed using streptavidin-conjugated horseradish peroxidase and the colorimetric substrate tetramethylbenzidine (TMB). The optical density was read at 450 nm using an ELISA plate reader (Tecan, Männedorf, Switzerland). Concentrations of galectin-9 in the plasma samples of the study participants on ART were determined by following the manufacturer’s instructions. The lower limit of detection (LLOD) for the assay was 93.8 pg/mL.

### 4.3. Full-Length Galectin-9 ELISA assay

Full-length galectin-9 concentrations in the plasma samples of the study participants were measured using the Human GAL9 ELISA Kit (Cat# HUES02158, ELISA Genie, Dublin, Ireland). The kit is designed to detect full-length galectin-9, as it uses a capture antibody against N-CRD and a detection antibody against C-CRD of galectin-9. The ELISAs were performed by strictly abiding by the protocols in the manufacturer’s manual after diluting HIV plasma samples to 1:10 with the sample diluent supplied with the kit. Full-length galectin-9 concentrations were determined by plotting a standard graph (four-parameter analysis) as per the instructions mentioned in the kit manual. The LLOD of full-length galectin-9 ELISA was 7.8 pg/mL.

### 4.4. Cystatin C ELISA Assay

Human cystatin C concentration in the plasma samples of the participants on ART was measured using the Human Cystatin C ELISA Kit (Cat# RD191009100, BioVendor, Brno, Czech Republic) as per the manufacturer’s instructions. Briefly, standards, quality control materials, and samples were first incubated in microtiter plates pre-coated with capture antibody. A secondary antibody conjugated with enzyme was added to the wells and incubated with the captured cystatin C after the recommended incubation and washing steps. Subsequently, the color was developed by the addition of substrate and stop solutions. A standard curve (four parameters) was constructed to determine the concentration of unknown samples. The sensitivity of the assay was 0.25 ng/mL.

### 4.5. Luminex Assays

Plasma levels of proinflammatory cytokines like IL-1β, IL-6, and IL-18 were assessed using a Bio-Plex Pro Human Cytokine Screening Panel (Bio-Rad, Hercules, CA, USA) as described previously [47]. The manufacturer’s instructions were followed while processing the plasma samples. The plate was read on a Bio-Plex 200 system (Bio-Rad, Hercules, CA, USA), and concentrations were determined based on the standard curves using Bio Plex Manager 6.0 software (Bio-Rad, Hercules, CA, USA).

### 4.6. Assays for Assessing HIV Reactivation

HIV reactivation was assessed by relative quantification of HIV-1 *gag* copies by RT PCR as well as by estimating intracellular P24-expressing CD4+ T cells by flow cytometry. Both of these assays were performed by stimulating PBMCs of HIV-infected individuals on ART for one year versus for more than 5 years’ duration with 350 nM of recombinant galectin-9 (R&D Laboratories, USA, accession number: NP002299) as described previously [6]. The cells were incubated with the recombinant protein at 37 °C for 24 h.

#### 4.6.1. Real-Time PCR

RNA extraction (Applied Biosystems, Waltham, MA, USA) was performed and cDNA synthesis (ThermoFisher Scientific, Waltham, MA, USA) was carried out to assess HIV-1 *gag* copies by real-time PCR assay on a 7500 HT FAST Real Time-PCR System (Applied Biosystems, Waltham, MA, USA). The β-actin gene was used as a housekeeping gene for calculating delta CT values for normalizing the expression of the *gag* gene. The expression index of the *gag* gene was calculated as 2-ΔCt ×1000 in unstimulated and stimulated PBMCs.

#### 4.6.2. Flow Cytometry

Surface staining was carried out for markers like CD3, CD8, and Tim-3 (using antibodies from BD Biosciences, Franklin Lakes, NJ, USA), followed by permeabilization and intracellular staining for FITC P24 (KC-57) (Beckman Coulter, Brea, CA, USA). The data analysis was completed by FlowJo version 8.0.3 after acquiring 50,000 gated events of lymphocytes on FACS Fusion I (BD Biosciences, USA).

### 4.7. Statistical Analysis

Statistical analysis was performed using GraphPad Prism version 9 software. Non-parametric tests were used for analysis of the data. The Spearman correlation was used for determining the coefficient of correlation, and *p*-values less than 0.05 were considered significant.

## 5. Conclusions

Among the two markers, total galectin-9 levels but not the full-length protein differed significantly between viremic and aviremic PLHIV treated with ART and correlated with markers of disease progression. Total galectin-9 levels were found to be affected by the duration of ART, decreasing its sensitivity to detecting viremia in PLHIV on long-term ART. However, the total galectin-9-to-CD4 count ratio was found to have the potential to detect viremic individuals irrespective of their treatment duration, with a sensitivity above 90% at a cutoff of 14.47 of the ratio. The specificity of the test was around 70%, suggesting the utility of the test as a screening test, and a need to perform the VL test for PLHIV with levels higher than the cutoff values was highlighted. Nonetheless, the test might have implications as a screening test in resource-limited conditions and for intermittent testing in addition to annual testing to rule out the presence of viremia. The combined effect of galectin-9 and cystatin C on HIV replication impacting HIV viremia and of its contribution to the development of non-AIDS events through inflammaging needs to be explored further. The results also warrant an urgent need to investigate inhibitors of galectin-9 to counter its effects of contributing to HIV viremia and non-AIDS events.

## Figures and Tables

**Figure 1 ijms-24-12273-f001:**
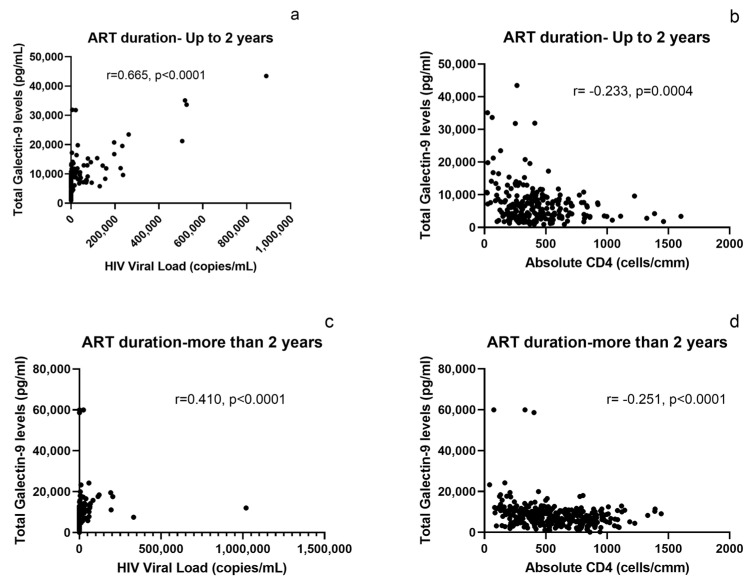
Correlation of plasma total galectin-9 levels with the markers of HIV disease progression in PLHIV on ART for up to and for more than 2 years: (**a**,**b**) The figures show correlation of plasma total galectin-9 levels with plasma viral load and CD4 cell counts, respectively, in PLHIV on ART for up to 2 years. (**c**,**d**) The graphs shown correlation of plasma total galectin-9 levels with plasma viral load and CD4 cell counts, respectively, in PLHIV on ART for more than 2 years. The correlation coefficient (r) and *p*-values as assessed by the Spearman test are mentioned in the figure.

**Figure 2 ijms-24-12273-f002:**
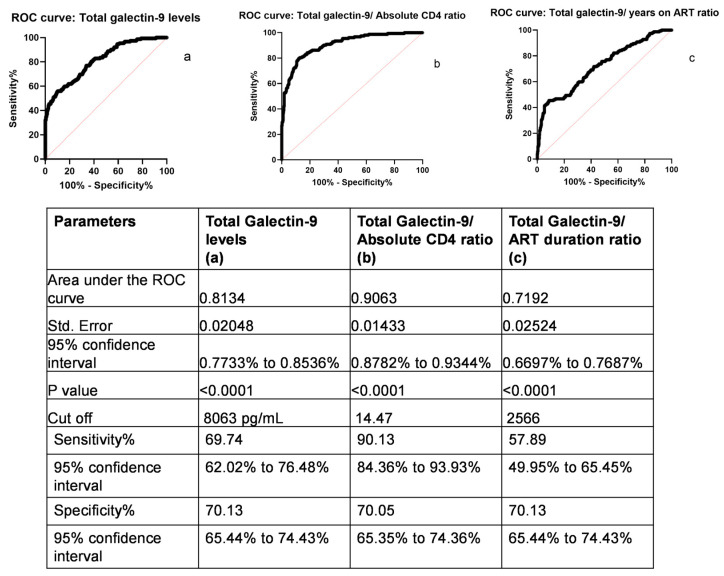
Receiver operating characteristic (ROC) curves for galectin-9 levels: ROC curves are plotted for total galectin-9 levels (**a**), the ratio of total galectin-9 levels to absolute CD4 count (**b**), and the ratio of total galectin-9 levels to duration of ART in years (**c**) for differentiating viremic and aviremic HIV-infected patients on ART. Sensitivity (%) and 100-specificity (%) are plotted on the Y and X axes, respectively. Black lines represent ROC and red lines represent reference lines for the curves. The figure also shows the AUC with standard errors and 95% confidence intervals (CI), *p*-values, cutoff points, sensitivities, and specificities with their respective CIs for all three ROC curves.

**Figure 3 ijms-24-12273-f003:**
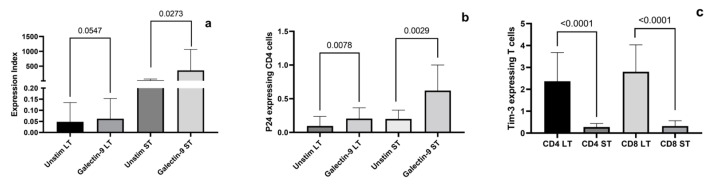
HIV reactivation induced by galectin-9 in PBMCs of HIV-infected individuals on long-term (LT) and short-term (ST) ART: HIV reactivation after stimulation with 350 nM recombinant galectin-9 was assessed by RT-qPCR and (**a**) the expression index of *gag* gene in unstimulated versus galectin-9-treated cells of individuals with long-term (LT) and short-term (ST) ART is shown. Flow cytometry assays were also conducted for detecting intracellular P24 expression. (**b**) The percentage of CD4 T cells expressing P24 in unstimulated versus galectin-9-treated cells of individuals with long-term (LT) and short-term (ST) ART is shown. (**c**) The expression of Tim-3 by CD4 and CD8+ T cells of individuals with long-term (LT) and short-term (ST) ART as assessed by flow cytometry is shown. Bars in the figure represent medians and error bars indicate interquartile ranges for the values. *p*-values showing significant differences (*p* < 0.05) between unstimulated and treated cells as calculated by the Wilcoxon signed-rank test are shown in the figure. *p*-values of unpaired data were calculated using the Mann–Whitney test.

**Table 1 ijms-24-12273-t001:** Characteristics of participants enrolled in the study.

Characteristics, Median (IQR)	Viremic PLHIV (*n* = 152)	Aviremic PLHIV (*n* = 395)	*p*-Value
Age, years	43.5 (35–49)	43 (37–49)	0.7342
Gender (M:F:TG)	76:76	174:220:1	0.2502
Years of ART	3 (1–7)	4 (1–8)	0.0582
VL, copies/mL	15,392 (3373–57,455)	Target not detected	<0.0001
CD4, %	15 (10–20)	27 (22–33.5)	<0.0001
Absolute CD4, cells/cmm	259.5 (167–402)	547 (388–771)	<0.0001
Total galectin-9, pg/mL	10,712 (7558–13,699)	5768 (3313–8689)	<0.0001
Full-length galectin-9, pg/mL	1365 (409–3266)	1257 (425–3193)	0.837

**Table 2 ijms-24-12273-t002:** Correlation analysis.

	Spearman r	95% Confidence Interval	*p* (Two-Tailed)
Total galectin-9 levels (pg/mL) vs. full-length galectin-9 (pg/mL)	0.2829	0.1091 to 0.4398	0.0013
Total galectin-9 (pg/mL) vs. VL (copies/mL)	0.5073	0.4403 to 0.5687	<0.0001
Total galectin-9 (pg/mL) vs. CD4 (%)	−0.2592	−0.3430 to −0.1712	<0.0001
Total galectin-9 (pg/mL) vs. absolute CD4 (cells/cmm)	−0.1863	−0.2683 to −0.1015	<0.0001
Total galectin-9 (pg/mL) vs. years of ART	0.2278	0.1443 to 0.3080	<0.0001
Total galectin-9 (pg/mL) vs. age (years)	0.0937	0.007518 to 0.1786	0.0283
Total galectin-9 (pg/mL) vs. IL-18 (pg/mL)	0.311	0.06321 to 0.5227	0.006
Full-length-to-total galectin-9 ratio vs. years of ART	0.3231	0.1169 to 0.5027	0.003
Total galectin-9 (pg/mL) vs. systolic blood pressure	0.2203	−0.02630 to 0.4416	0.0355 (one-tailed)
VL (copies/mL) vs. CD4 (%)	−0.534	−0.5967 to −0.4648	<0.0001
VL (copies/mL) vs. absolute CD4 (cells/cmm)	−0.5064	−0.5680 to −0.4393	<0.0001
VL (copies/mL) vs. years of ART	−0.09152	−0.1764 to −0.005260	0.0323
VL (copies/mL) vs. galectin-9 (pg/mL)-to-absolute CD4 (cells/cmm) ratio	0.6422	0.5885 to 0.6903	<0.0001
Cystatin C levels (pg/mL) vs. total galectin-9 (pg/mL)	0.239	0.03582 to 0.4236	0.0183
Cystatin C levels (pg/mL) vs. age (years)	0.209	0.003823 to 0.3970	0.04
Cystatin C levels (pg/mL) vs. years of ART	0.308	0.1095 to 0.4825	0.002
Cystatin C levels (pg/mL) vs. serum creatinine (mg/dL)	0.3277	0.09530 to 0.5263	0.0053
Cystatin C levels (pg/mL) vs. blood urea nitrogen (mg/dL)	0.276	0.03702 to 0.4854	0.0206

## Data Availability

Data will be made available upon request.

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
