# Peer review of "Antiretroviral Treatment-Induced Galectin-9 Might Impact HIV Viremia in Addition to Contributing to Inflammaging"

_ijms, 2023, doi:10.3390/ijms241512273_

Round 1

Reviewer 1 Report

In this article, Ashwini Shete and colleagues investigate levels of Galectin-9 in HIV-1 infected individuals and correlate with the levels of HIV-1 viremia and the years on treatment. The authors observed higher levels of Galectin-9 during the early stages of infection in the Cart treated patients. The Galectin-9 levels correlated with viremia levels and inversely with CD4 counts. ROC analysis showed the best prognostic value for Galectin-9/CD4 ratio. The authors also test Galectin-9 for induction of HIV-1 in latently infected cells. Overall, this is a well written paper. However, it has many missing points. The cohort does not have HIV- negative participants. Thus, it is unclear what are the levels of galectin-9 in the non-infected individuals. The activation of HIv1- by Galectin-9 is quite modest, and there is no positive control to compare with. The title implies inflammation analysis, which was not done. Finally, it is hard to believe that double testing of Galctin-9 and CD4 could improve the diagnostics of HIV-1 infection that is currently done by antibodies-based tests and can be conducted in the poor resources settings. All these considerations (outlined below) reduce the reviewer’s enthusiasm.

Major critiques:

1.      This study only investigates HIV-1+ individuals. Controls without HIV-1 infection need to be tested to prove that Galectin-9/CD4 ratio has validity as diagnostics marker.

2.      Activation of HIV-1 infected cells by Galectin-9 lacks control, i.e., TCR receptor activation. It seems that the activation is quite small and its significance might be overstated.

3.      Galectin-9 was mentioned in the relation to aging, but no attempt was made to correlate its levels based on age.

4.      Unclear if levels of Calectin-9 reflect inflammation and whether there is a correlation with the inflammation (i.e., IL-1beat, IL-6 or IL-18 levels).   

5.      I do not see a clear explanation of why total and full Galectin-9 assays show different results. Further analysis is needed to demonstrate that indeed the assays are specific, as this is only ELISA kits.

6.      The reviewer is not convinced that Galectin-9 ELISA and CD4 counts analysis by a CBC instrument is superior to the simple saliva HIV-1 positivity test.

7.      Galectin-9 levels were previously shown to correlate with long-term morbidity in the gaining HIV-1+ population. It is unclear who the data from this study that showed reduction of Galectin-9 after 2 years of treatment relate to this previous observation.

Minor corrections:

8.      Lane 67, “…data of PLHIV on ART… was not studied…” should be “…data of PLHIV on ART… were not studied…”

Author Response

Reviewer 1:

In this article, Ashwini Shete and colleagues investigate levels of Galectin-9 in HIV-1 infected individuals and correlate with the levels of HIV-1 viremia and the years on treatment. The authors observed higher levels of Galectin-9 during the early stages of infection in the Cart treated patients. The Galectin-9 levels correlated with viremia levels and inversely with CD4 counts. ROC analysis showed the best prognostic value for Galectin-9/CD4 ratio. The authors also test Galectin-9 for induction of HIV-1 in latently infected cells. Overall, this is a well written paper. However, it has many missing points. The cohort does not have HIV- negative participants. Thus, it is unclear what are the levels of galectin-9 in the non-infected individuals. The activation of HIv1- by Galectin-9 is quite modest, and there is no positive control to compare with. The title implies inflammation analysis, which was not done. Finally, it is hard to believe that double testing of Galctin-9 and CD4 could improve the diagnostics of HIV-1 infection that is currently done by antibodies-based tests and can be conducted in the poor resources settings. All these considerations (outlined below) reduce the reviewer’s enthusiasm.

Response:

 We would like to thank the reviewer for summarizing the findings and giving very constructing suggestions. Concerns raised by the reviewer are addressed below against each critique. Major focus of the manuscript is on determining association of Galectin-9 levels with HIV viremia. We propose estimating the levels and their ratio with CD4 count for identifying viremic individuals who are already diagnosed with HIV and put on antiretroviral therapy. We do not propose this test for diagnosis of HIV infection for which antibody based tests are being used. The test is primarily proposed as a screening test for viral load testing which is expensive. Individuals having the ratio above the cut-off level need to undergo viral load testing to confirm their viremic status. Since we reported the sensitivity of the test to be more than 90%, individuals with the ratio below cut-off may be excluded from frequent viral load testing to save the cost in resource poor countries. We have explained this against each point and also in the revised manuscript to avoid any confusion.

Major critiques:

  1. This study only investigates HIV-1+ individuals. Controls without HIV-1 infection need to be tested to prove that Galectin-9/CD4 ratio has validity as diagnostics marker.

Response:

As mentioned above, the study proposes the use of Galectin-9/CD4 ratio as a prognostic marker and not as a diagnostic marker. We are sorry for not mentioning it explicitly.  It is now clarified in introduction and discussion parts. (Page Nos. 2,7)

Since the ratio is not proposed for diagnosis of HIV infection but is to monitor patients on ART for virologic failure, cases in the study were the patients with viremia whereas the controls were aviremic HIV infected individuals on ART. However, we agree that it is preferable to add the data on HIV uninfected individuals. We have estimated the levels and ratio in HIV uninfected individuals as a part of a different study which is now added in the manuscript to address the concerns raised by the reviewer (Page nos.3,5,10). Author list is also revised to add one more author who was involved in data collection for the study.

  1. Activation of HIV-1 infected cells by Galectin-9 lacks control, i.e., TCR receptor activation. It seems that the activation is quite small and its significance might be overstated.

Response:

We performed the assays on PBMCs of HIV infected individuals on virally suppressive ART for short and long term durations. Circulating HIV reservoir diminishes in patients on virally suppressive ART and hence it is likely that HIV reactivation observed after stimulation of PBMCs from these patients was quite small. We have been performing the ex vivo assays in patients’ PBMCs using a variety of latency reversing agents. We did try reactivation using anti CD3/CD28 antibodies or PMA/Ionomycin for TCR activation in past, but did not observe much increase in P24 expression in the ex vivo assays. Hence, we did not use these controls to save on PBMCs which were required to be used for flow as well as real time assays. However we confirmed the gag expression in PBMCs by flow as well as real time assays which showed increased expression suggesting HIV reactivation by Galectin-9. We have explained this in the discussion part of the revised manuscript. (Page Nos. 8,9)

  1. Galectin-9 was mentioned in the relation to aging, but no attempt was made to correlate its levels based on age.

Response:

We would like to thank the reviewer for suggesting this. The levels showed a weak positive correlation with age. Results and discussion related to the same is now added to the revised manuscript. (Page Nos. 3,9 and Table 2)

  1. Unclear if levels of Calectin-9 reflect inflammation and whether there is a correlation with the inflammation (i.e., IL-1beat, IL-6 or IL-18 levels).

Response:

We would again like to thank the reviewer for suggesting adding the data on correlation of Galectin-9 levels with other inflammatory markers. Results of correlation analysis with IL-1beta, IL-6 or IL-18 levels are now added. The levels did not show any correlation with IL-1b (r=-0.024, p=0.425) and IL-6 (r=0.003, p=0.489), however, positive correlation of the levels with IL-18 (r=0.311, p=0.006) was observed. IL-18 has been shown to participate in processes involved in inflammaging and its raised levels have been reported in individuals with metabolic syndrome.

Methodology, result and discussion parts are now revised to add these findings. (Page Nos. 6,9,11 and Table 2)

  1. I do not see a clear explanation of why total and full Galectin-9 assays show different results. Further analysis is needed to demonstrate that indeed the assays are specific, as this is only ELISA kits.

Response:

Both the kits used in the study were commercially available with certificate of analysis supplied with the kits. Both the kits were based on sandwich ELISA which uses two different antibodies for detection of an antigen and hence are known to be highly specific. Their specificity has been confirmed using other Galectins and Galectin-9 from other species as mentioned on the respective kit inserts.

The difference in the results could be because of the differences reported in functionality of full length versus fractions of the Galectin-9 proteins and their relative proportion in viremic and aviremic individuals. Full length Galectin-9 levels increased after long term ART even in aviremic individuals possibly suggesting its induction by antiretroviral drugs. Viremic individuals are more likely to have higher level of immune activation contributing to inflammatory responses leading to more cleavage of the protein by various inflammatory enzymes. The cleavage might lead to higher total Galectin-9 levels decreasing the full length Galectin-9 levels. This is now discussed in the revised manuscript. (Page no. 8)

  1. The reviewer is not convinced that Galectin-9 ELISA and CD4 counts analysis by a CBC instrument is superior to the simple saliva HIV-1 positivity test.

Response:

As mentioned in response to the first comment, Galectin-9/CD4 ratio is proposed here as a prognostic marker and not as a diagnostic marker. It might be used to monitor virologic failure once HIV diagnosis is done and the patients are put on antiretroviral therapy. So it will not replace blood/ saliva based HIV tests. But it can be used as a supplementary test for viral load testing in already diagnosed patients accessing antiretroviral treatment. The combination is proposed as a screening low cost test for the resource poor settings and confirmation has to be done by viral load test.   

It is now clarified in introduction and discussion parts. (Page Nos. 2,7)

  1. Galectin-9 levels were previously shown to correlate with long-term morbidity in the gaining HIV-1+ population. It is unclear who the data from this study that showed reduction of Galectin-9 after 2 years of treatment relate to this previous observation.

Response:

Galectin-9 levels did not reduce after 2 years in our study. In fact they showed positive correlation with ART duration increasing steadily with the duration. However, their correlation with HIV viral load was weaker in individuals on ART for more than 2 years than those on ART for less than 2 years. This is now mentioned clearly in the manuscript. (Page no.3 and Table 2)

Minor corrections:

  1. Lane 67, “…data of PLHIV on ART… was not studied…” should be “…data of PLHIV on ART… were not studied…”

Response:

We are sorry for the grammatical mistake. It is now corrected. Page no.2)

Reviewer 2 Report

In this study, the authors assessed plasma galectin-9 levels in people living with HIV (PLHIV) on antiretroviral therapy (ART) and their associations with plasma viral load, Cystatin C levels as well as other parameters possibly contributing to non-AIDS events. The authors concluded that among the two markers, a total Galectin-9 levels, but not the full length protein, differed significantly between the viremic and aviremic PLHIV and correlated with markers of the disease progression.

Comments

This is an interesting study. The reviewer has some concerns as follows:

1. Although the statistical analysis for the most of parameters is significant (p<0.05), the correlation between the two variables is not high or is weak, such as “Total Galectin -9 levels (pg/mL) Vs full length Galectin-9 (pg/mL)” Spearman r = 0.2829; “Total Galectin -9 (pg/mL) Vs Absolute CD4 (cells/cmm)” Spearman r = -0.1863. The authors need to explain and discuss this issue.

2. The spelled-out version for the abbreviations of PLHIV and ART need to be shown when they first appear.

3. In the Materials and Methods, the catalog numbers for all ELISA kits obtained from various manufacturers can be added.

4. In line 304, the spelled-out version for TND needs to be shown (is Target not detected?).

5. The limitation(s) of this study can be described in the end of Discussion section.

Author Response

Reviewer 2:

In this study, the authors assessed plasma galectin-9 levels in people living with HIV (PLHIV) on antiretroviral therapy (ART) and their associations with plasma viral load, Cystatin C levels as well as other parameters possibly contributing to non-AIDS events. The authors concluded that among the two markers, a total Galectin-9 levels, but not the full length protein, differed significantly between the viremic and aviremic PLHIV and correlated with markers of the disease progression.

We would like to thank the reviewer for summarizing the findings and giving invaluable suggestions. Responses to the comments are as given below:

Comments

This is an interesting study. The reviewer has some concerns as follows:

  1. Although the statistical analysis for the most of parameters is significant (p<0.05), the correlation between the two variables is not high or is weak, such as “Total Galectin -9 levels (pg/mL) Vs full length Galectin-9 (pg/mL)” Spearman r = 0.2829; “Total Galectin -9 (pg/mL) Vs Absolute CD4 (cells/cmm)” Spearman r = -0.1863. The authors need to explain and discuss this issue.

Response:

Weak association between full length and total galectin-9 could be because of their differential relative proportion in viremic and aviremic individuals. Full length Galectin-9 levels increased after long term ART even in aviremic individuals possibly suggesting its induction by antiretroviral drugs. Viremic individuals are more likely to have higher level of immune activation contributing to inflammatory responses leading to more cleavage of the protein by various inflammatory enzymes. The cleavage might lead to higher total Galectin-9 levels decreasing the full length Galectin-9 levels. This is discussed in the revised manuscript. (Page no.8)

The levels were found to correlate positively with duration of ART even in aviremic individuals suggesting involvement of factors other than HIV viremia playing role in its induction and vice a versa weakening their association. Hence although strong association has been reported with CD4 counts in patients with AIDS, it tends to become weak after ART initiation. There has been a report on mixed HIV infected population consisting of controllers, non-controllers and ART suppressed individuals where no association is observed. This is discussed in the revised manuscript. (Page no.7)

Weaker associations between other parameters are explained while discussing limitations of the study in the discussion section (Page Nos.9-10).

  1. The spelled-out version for the abbreviations of PLHIV and ART need to be shown when they first appear.

Response:

Sorry for missing this. They are expanded now where they appeared first. (Page no.2)

  1. In the Materials and Methods, the catalog numbers for all ELISA kits obtained from various manufacturers can be added.

Response:

Catalogue numbers for all ELISA kits used for the study are now added in the methodology part. (Page no.10)

  1. In line 304, the spelled-out version for TND needs to be shown (is Target not detected?).

Response:

TND is now spelled out. (Page no.10)

  1. The limitation(s) of this study can be described in the end of Discussion section.

Response:

Limitations of the study are now described in the discussion session. (Page No. 8,9-10)
